# A nonstructural protein 1 capture enzyme-linked immunosorbent assay specific for dengue viruses

Pei-Yin Lim[1¤a], Appanna Ramapraba[1¤b], Thomas Loy[1¤c], Angeline Rouers[1¤c], Tun-Linn Thein[2], Yee-Sin Leo[2], Dennis R. Burton[3], Katja Fink[1¤d], Cheng-I Wang[1]*

1 Singapore Immunology Network, Agency for Science, Technology and Research, Singapore, Singapore,
2 National Centre for Infectious Diseases, Singapore, Singapore, 3 Department of Immunology and Microbiology, The Scripps Research Institute, La Jolla, CA, United States of America

¤a Current address: Hilleman Laboratories Pte Ltd, Singapore, Singapore
¤b Current address: Department of Medicine, Surgery & Dentistry, University of Salerno, Baronissi (Salerno), Italy
¤c Current address: A*STAR Infectious Disease Labs, Agency for Science, Technology and Research, Singapore, Singapore
¤d Current address: ImmunoScape Pte Ltd, Singapore, Singapore
* Wang_ChengI@immunol.a-star.edu.sg

**Data Availability Statement:** All relevant data are within the paper and its Supporting Information files.

## Abstract

Dengue non-structural protein (NS1) is an important diagnostic marker during the acute phase of infection. Because NS1 is partially conserved across the flaviviruses, a highly specific DENV NS-1 diagnostic test is needed to differentiate dengue infection from Zika virus (ZIKV) infection. In this study, we characterized three newly isolated antibodies against NS1 (A2, D6 and D8) from a dengue-infected patient and a previously published human anti-NS1 antibody (Den3). All four antibodies recognized multimeric forms of NS1 from different serotypes. A2 bound to NS1 from DENV-1, -2, and -3, D6 bound to NS1 from DENV-1, -2, and -4, and D8 and Den3 interacted with NS1 from all four dengue serotypes. Using a competition ELISA, we found that A2 and D6 bound to overlapping epitopes on NS1 whereas D8 recognized an epitope distinct from A2 and D6. In addition, we developed a capture ELISA that specifically detected NS1 from dengue viruses, but not ZIKV, using Den3 as the capture antibody and D8 as the detecting antibody. This assay detected NS1 from all the tested dengue virus strains and dengue-infected patients. In conclusion, we established a dengue-specific capture ELISA using human antibodies against NS1. This assay has the potential to be developed as a point-of-care diagnostic tool.

## Introduction

Dengue is a global health problem affecting 129 countries, with 3.9 billion people at risk of infections. In 2019, 5.2 million cases of dengue infection were reported to WHO [1]. Infection is caused by the dengue virus (DENV), a mosquito-borne flavivirus consisting of four serotypes (DENV-1 to -4). Infection causes a spectrum of diseases, ranging from asymptomatic, flu-like symptoms to fatal complications involving increased vascular permeability that could

**Funding:** This work was supported by SIgN Core grant. The funders had no role in study design, data collection and analysis, decision to publish, or preparation of the manuscript.

**Competing interests:** The authors have declared that no competing interests exist.

result in fluid loss and organ dysfunction. Other mosquito-borne flaviviruses that are closely related to DENV and can cause serious disease include Zika virus (ZIKV), Yellow fever virus, West Nile virus, and Japanese encephalitis virus.

Dengue non-structural protein-1 (NS1) is a 48–50 kDa protein that is 40–70% conserved across the flaviviruses [2]. NS1 is translated from the viral RNA as part of a polyprotein, cleaved into monomers, glycosylated, and then dimerized. In mammalian cells, the dimeric form of NS1 is further processed and then secreted from infected cells as soluble hexamers and potentially other multimers [3]. During the acute phase of infection, NS1 is an important diagnostic marker, especially for the use at point-of-care, because it can be detected in the blood (reviewed by [4]). NS1 has also been shown to play important roles during virus pathogenesis. Severe dengue illness has been associated with high levels of circulating NS1 in the plasma of dengue infected patients [5, 6]. *In vitro* and animal studies suggest that DENV NS1 causes plasma leakage through disrupting the tight junction or inducing glycocalyx degradations (reviewed by [7]). In addition, antibodies against NS1 have been shown to cross react with host proteins through molecular mimicry, leading to tissue damage, thrombocytopenia, and coagulopathy (reviewed by [8]).

Enzyme-linked immunoassays to detect the presence of dengue NS1 in the blood are commercially available as dengue diagnostic tests and the performance of these assays has been tested by various groups. Depending on the study, the reported sensitivity of these test kits ranges from ~30% to 90%, depending on the serotype of infection [9–12]. In addition, the emergence of ZIKV, another flavivirus, in regions endemic for DENV infection, such as South America and South East Asia [13], has made the detection of DENV NS1 a challenge. Due to the sequence similarity of NS1 from ZIKV to all four serotypes of dengue viruses, the primary concern of using these diagnostic assays is an inability to differentiate between ZIKV and DENV infections. A highly specific dengue diagnostic test is therefore needed.

The goal of this study is to develop an NS1 capture enzyme-linked immunoassay that can positively detect all four dengue serotypes whilst providing a negative signal in the presence of Zika virus.

## Methods and materials

### Ethics statement

All studies involving human subjects have been reviewed by Institutional Review Board of Singapore National Healthcare Group Ethical Domain. Written informed consent was obtained from all participating human subjects and the data was fully anonymized.

### Sorting of NS1-specific B cells from DENV-positive patient

The study was approved by the Institutional Review Board of Singapore National Healthcare Group Ethical Domain (DSRB Ref: B/05/013). Written informed consent was obtained from the patient. Whole blood from a DENV-positive patient at convalescence phase (3–4 week after fever onset) was collected into CPT tubes (Becton Dickinson). The patient was confirmed DENV positive by DENV-specific RT-PCR, Panbio Dengue IgG Indirect ELISA and Panbio Dengue IgM capture ELISA (Inverness Medical, Queensland, Australia). Peripheral blood mononuclear cells (PBMCs) were isolated from whole blood by CPT tube (Becton Dickinson) purification and stored at -80˚C until used. Thawed PBMC were labeled with antibodies against CD3-BV450 (BD Horizon, Clone#: UCHT1), CD19-BV605 (Pharmingen, Clone#: SJ25C1), CD27-PE (Pharmingen, Clone#: M-T271), CD45-V500 (BD Horizon, Clone#: H130), IgD-PECy7 (Biolegend, Clone#: IA6-2), and fluorescently labelled recombinant DENV2-NS1 antigen and DENV4-NS1 antigen (The Native Antigen Company). Recombinant

NS1 was labeled in house with Alexa Fluor 488 (Alexa Fluor 488 Antibody labeling kit, ThermoFisher Scientific) and Alexa Fluor 647 (Alexa Fluor 647 Antibody Labeling kit, ThermoFisher Scientific), respectively following the manufacturer's instruction. PBMCs were resuspended in sorting buffer (PBS, 2% FBS, 2 mM EDTA) after incubation with antibodies and antigen. Memory B cells that stained positive for DENV-2-NS1 and DENV-4-NS1 were sorted into 96-well PCR plates containing 10 μl/well of 10 mM Tris-HCL with 40 U/μl RNase inhibitor (Promega), placed on dry ice immediately, and then stored at -80˚C.

## Ig cloning, expression, purification, and sequencing

Methods to clone and express the antibodies have been described previously [14]. Briefly, the human IgG heavy and light chains were amplified from mRNA of single B cells using One-step RT-PCR (Qiagen), the resulting RT-PCR products were used for nested PCR with primers including restriction sites. The PCR products were cloned into the pTT5 expression vector (National Research Council of Canada). The plasmids expressing the heavy and light chain (IgG1 format) were co-transfected into HEK293-6E cells using 293fectin™ (Thermo Fisher Scientific) as per manufacturer's instructions with the following modifications. The transfected cells were cultured in FreeStyle F17 Expression Medium (ThermoFisher Scientific) supplemented with 0.5% (w/v) Tryptone N1. The supernatant was harvested 5 days post-transfection, and the antibodies were purified using Protein G beads (Merck Millipore).

The PCR products of the heavy and light chain variable regions from single B cells were sequenced by Sanger sequencing (1st Base, Singapore). The quality of the sequences was checked using CodonCode Aligner software, and the sequences were trimmed and further analyzed using the IMGT database (http://www.imgt.org/).

## Production and purification of Den3 antibody

Den3 antibodies were expressed in Chinese hamster ovary cells in glutamine-free custom formulated Glasgow minimum essential medium (MediaTech Cellgro) and purified using Protein A beads (Merck Millipore) as previously described [15, 16].

## Selection of NS1-specific antibodies by indirect ELISA

Recombinant NS1 proteins from DENV-1-Nauru/Western Pacific/1974 (Accession: M23027.1), DENV-2-Thailand/16681/84 (U87411.1), DENV-3-Sri Lanka D3/H/IMTSSA-SRI/2000/1266 (AY099336.1), DENV-4-Dominica/814669/1981 (AF326573.1), ZIKV-Suriname (AZS35340.1), and ZIKV-Uganda MR766 (AWF93629.1) produced in HEK293 cells were purchased from The Native Antigen Company (United Kingdom). Nunc Maxisorp plates (ThermoFisher Scientific) were coated with recombinant NS1 (50 μL/well of 2 μg/mL) in coating buffer (0.1M NaHCO₃) overnight at 4˚C. Wells were washed three times with PBST (PBS supplemented with 0.05% Tween) and blocked with PBS supplemented with 5% skim milk (Sigma-Aldrich) for 1 h at RT. After washing the wells once with PBST, purified antibodies (50 μL of 1μg/mL) was added into the wells for 2 h at RT. Wells were washed three times with PBST, and 50 μL/well of peroxidase-conjugated anti-human IgG (1/5000 dilution, Sigma Aldrich) or HRP-conjugated anti-rabbit IgG (H+L) (1/5000 dilution, Promega) was added for 2 h at RT. After the wells were washed 3X with PBST, 3,3',5,5'-Tetramethylbenzidine (TMB) Liquid Substrate system for ELISA (50 μL/well, Sigma) was added for ~ 5minutes, and then 1M HCL was added. The OD of the wells were determined at 450nm and 650nm (reference OD).

## Selection of antibody pairs for NS1-capture ELISA

Nunc Maxisorp plates (ThermoFisher Scientific) were coated with purified antibodies at 1 μg/mL in coating buffer overnight at 4°C. Wells were washed three times with PBST and blocked with PBS supplemented with 5% skim milk (Sigma-Aldrich). After 1 h at RT, skim milk solution was removed and recombinant NS1 (50 μL of 2μg/mL) was added into the wells for 2 h at RT. Wells were washed three times with PBST, and 50 μL/well of peroxidase-conjugated purified antibodies (1/200 dilution, Sigma Aldrich) was added for 2h at RT. After the wells were washed 3X with PBST, TMB Liquid Substrate system for ELISA (50 μL/well, Sigma) was added for ~ 5minutes, and then 1M HCL was added. The OD of the wells were determined at 450nm and 650nm (reference OD). Purified antibodies were conjugated with peroxidase using the EZ-Link plus activated peroxidase kit (ThermoFisher Scientific) as per manufacturer's instructions.

## Evaluation of the performance of the NS1-capture ELISA

NS1-capture ELISA was performed as described above using Den3 [16] as the capture antibodies and D8-HRP as the detecting antibodies. To determine if the NS1-capture ELISA could detect NS1 from different dengue strains/isolated, supernatant from Vero cells infected with different dengue viruses was added into the wells coated with Den3 antibodies. To determine the limit of detection of the assay, various concentrations of purified recombinant NS1 prepared in human plasma from dengue-negative patient were added into the Den3 coated wells and then the NS1-capture ELISA was performed as described above using D8-HRP as the detecting antibodies. The limit of detection (LOD) of the assay was defined as the concentration of NS1 with an OD value that was equal to the 2-fold the average OD of wells containing no NS1.

To detect NS1 from dengue-infected patient samples, NS1-capture ELISA was performed as described above, except that human plasma was added into the wells coated with antibodies, instead of purified recombinant NS1. Clinical specimen was considered positive for NS1 when its OD reading was greater than 2 times the OD of the negative plasma control in each plate [17].

## SDS-PAGE and Western blot analysis

Recombinant NS1 (final concentration at 0.03 μg/μL for DENV-1, -3, and -4, and 0.01 μg/μL for DENV-2) were prepared in PBS and 4X Laemmli Sample Buffer (Bio-Rad). The samples were divided into 2 tubes: (1) untreated and (2) incubated at 90°C for 10 min. The proteins were separated on an Any kD™ Mini-PROTEAN® TGX Stain-Free™ Protein Gels (Bio-Rad) and then transferred onto a Hybond PVDF membrane (0.45μm, Amersham). The membrane was incubated in PBS supplemented with 5% (w/v) skim milk overnight at 4°C, then purified antibodies at 1μg/mL for 2 h at RT. The membranes were washed 3 times with PBST, 5 minutes each, then peroxidase-conjugated anti-human IgG (1/10,000 dilution, Sigma Aldrich) was added for 1 h at RT. The membranes were washed three times with PBST, chemiluminescent substrates (WesternBright Sirius chemiluminescent Detection kit, Advansta) were added and then developed using BioRad ChemiDoc system. For SDS-PAGE, the gel was stained with InstantBlue™ (Expedeon).

## Biolayer light interferometry (BLI) analysis

Analysis of binding of antibodies to recombinant NS1 was performed using an Octet RED96 instrument (ForteBio; Pall Life Sciences) as described previously [18]. Purified antibodies A2, D6, D8, or Den3 (10μg/mL) were loaded onto anti-human IgG Fc Capture (AHC) Biosensors

(ForteBio) for 200 sec. The biosensor tips were washed with binding buffer [PBS supplemented with 0.1% Tween and 0.1% (w/v) BSA] for 200 sec, then immersed into different concentrations of recombinant NS1 (0, 1.562, 3.125, 6.25, 12.5, 25, 50, 100 nM) for 200 sec (association), followed by a subsequent immersion in PBS for 600 sec (dissociation). Octet® Data Acquisition software was used for data acquisition, and Octet® Data Analysis software was used for affinity calculations. Briefly, the background signal for each antibody was the signal from the control sensors that were loaded with the specific antibody and incubated with 0 nM of recombinant NS1. Background signal was subtracted from all other results (biosensors loaded with antibodies, incubated with recombinant NS1 at 1.562, 3.125, 6.25, 12.5, 25, 50, 100 nM), the Y-axis was aligned to the association step (199.8 second), and then the association and dissociation constants were analyzed using the 1:1 model with global fitting. New biosensor tips were used for all experiments.

## Preparation of culture supernatant containing NS1 secreted from infected cells

Culture supernatants containing NS1 were prepared in Vero cells that were maintained in RPMI (Hyclone™) supplemented with 5% heat-inactivated Fetal bovine serum (HI-FBS, Gibco) at 37°C, 5%$CO_2$. The virus strains used in this study included DENV-1-WestPac (accession no: U88535.1) [19], DENV-1-08K3126 [14], DENV-2-TSV01 (AY037116.1) [20], DENV-2-D2Y98P derived from an infectious clone (JF327392.1) [21], DENV-3-VN32/96 (EU482459), DENV-3-CHD94-089, DENV-4-2641Y08 (HQ875339.1), DENV-4-TVP360 (GU289913.1) [22], and ZIKV(PLCal) (KX694532.2). Monolayers of Vero cells in T75 flasks were inoculated with $10^5$ focus forming units of virus for 1 h at 37°C, 5%$CO_2$. Viral inoculum was removed, fresh RPMI supplemented with 5% HI-FBS was added, and culture was incubated at 37°C, 5%$CO_2$. Culture supernatant was harvested when >50% cytopathic effect was observed, clarified by centrifugation at 2,000xg for 5 minutes, concentrated using a 30kDa Amicon® Ultra-15 centrifugal filter unit (MerkMillipore), and stored at -80°C.

## Patient cohort for ELISA validation

All patients were recruited following IRB approval from National Healthcare Group domain specific Review Board (NHG DSRB Ref: 2015/0528 and NHG DSRB Ref: 2016/00982). The details of these patients have been previously described [23]. Patients were enrolled into the study after obtaining their consent, and all data was fully anonymized.

## Results

### Isolation and characterization of DENV NS1-specific antibodies

To isolate antibodies that recognized NS1 from multiple serotypes, we isolated PBMC from a DENV positive patient, incubated the PBMC with antibodies and recombinant NS1, and then sorted memory B cells (CD19+CD27+IgD−) recognizing NS1 of the DENV-2 alone or DENV-2 and DENV-4 from a convalescent DENV-2 patient at one cell per well. The gating strategy of the memory B cells was shown in S1 Fig. The variable regions of the heavy and light chains from these cells were reverse transcribed from the mRNA and amplified by PCR. The PCR products were cloned into the expression vectors. Following the transient expression and purification, the antibodies were tested for their ability to bind a multimeric form of recombinant NS1 from all four dengue serotypes using an indirect ELISA. Through our screening process, we identified three antibodies A2, D6, and D8 that bound to NS1 from multiple dengue serotypes. Specifically, A2 interacted with NS1 from DENV-1, -2, and -3, D6 interacted with NS1

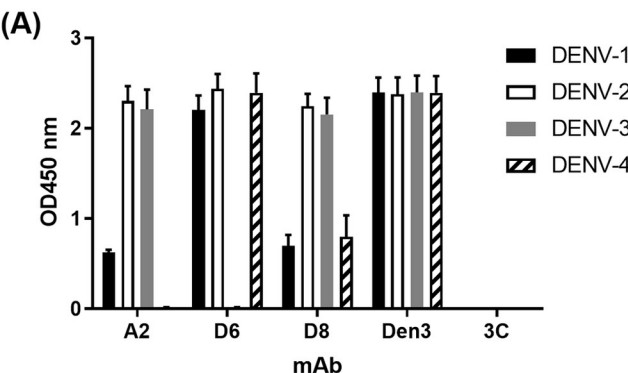

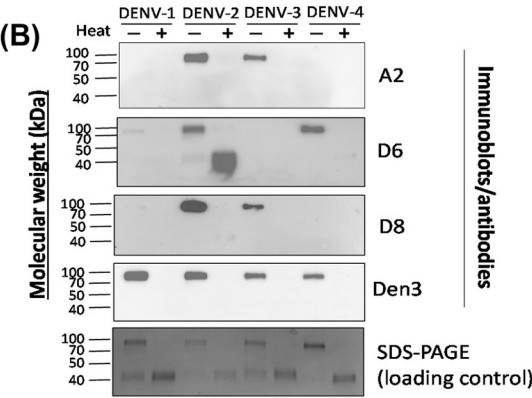

**Fig 1. Human antibodies against NS1 recognize multimeric forms of DENV NS1.** (A) Indirect ELISA demonstrating the specificity of the antibodies. Wells were coated with NS1 from various dengue viruses, blocked, and then purified antibodies were added. The bound antibodies were detected by the addition of HRP-conjugated anti-human IgG followed by TMB substrate. Average OD values ± SEM from 2 experiments, performed in duplicates are shown. (B) Immunoblot analysis. Purified NS1 from various dengue viruses were untreated or incubated at 90°C for 10 minutes, separated on an SDS-PAGE, and then transferred onto a PVDF membrane. The membranes were incubated with the purified antibodies, followed by a HRP-conjugated anti-human IgG and then chemiluminescent substrate. As a loading control, the SDS-PAGE was stained with InstantBlue^TM.

from DENV-1, -2, and -4, and D8 interacted with NS1 from all four dengue serotypes. The positive control Den3, a dengue virus NS1-specific human IgG1 antibody [16], bound to all four dengue serotypes. Finally, the negative control, 3C, a DENV Envelope-specific human IgG1 antibody [24], did not interact with any of the NS1 proteins (Fig 1A).

To test whether the monomer or dimer forms of NS1 was recognized by these antibodies, we performed an immunoblot analysis using untreated NS1 and NS1 that was heated at 90°C for 10 minutes. Consistent with previously published data [25], treatment of NS1 at 90°C dissociated dimers into monomers. All four antibodies recognized the dimeric form of NS1 from various dengue serotypes. Specifically, A2 and D8 bound to dimeric forms of NS1 from DENV-2 and DENV-3, and D6 to DENV-2 and DENV-4. D6 also bound to monomeric form of NS1 from DENV-2. The loss of detection of the DENV-1 NS1 by A2 and D6, and the loss of detection of the DENV-1 NS1 and DENV-4 NS1 by D8 by immunoblot analysis were possibly due to the lower affinity of the antibodies towards these serotypes, or the requirement of non-denatured protein for binding. Consistent with results from indirect ELISA, the positive control Den3 bound to NS1 dimer from all four DENV serotypes (Fig 1B).

Finally, BLI analysis was performed to examine the Ab binding kinetics and affinity for the commercially available, presumably hexameric, recombinant NS1. Briefly, antibodies were immobilized onto the biosensors, the antibody-loaded biosensors were immersed into solutions containing various concentrations of recombinant NS1 to allow for NS1 to bind to the biosensors, and then NS1-bound biosensors were immersed in buffer to allow for NS1 to dissociate from the antibodies. The association and dissociation constants were calculated using the 1:1 model with global fitting using all concentrations of NS1, and the mean $K_D$ values from 2 independent experiments were shown in Fig 2. The serotype-preference of NS1 binding to the immobilized antibodies was similar to that observed from the indirect ELISA. A2 bound to NS1 from DENV-2 and DENV-3 with higher affinities, and DENV-1 and DENV-4 with lower affinities. D6 bound to NS1 from DENV-1, -2, and -4, and D8 and Den3 bound to NS1 from all four serotypes, albeit with different affinity (Fig 2). A slight upward drifting during the dissociation phase was observed for antibodies with high affinities towards the NS1 (KD $<10^{-12}$ M). The reason for this upward drifting could not be ascertained experimentally. It could theoretically involve a conformational change of Ab-NS1 complex during the dissociation phase.

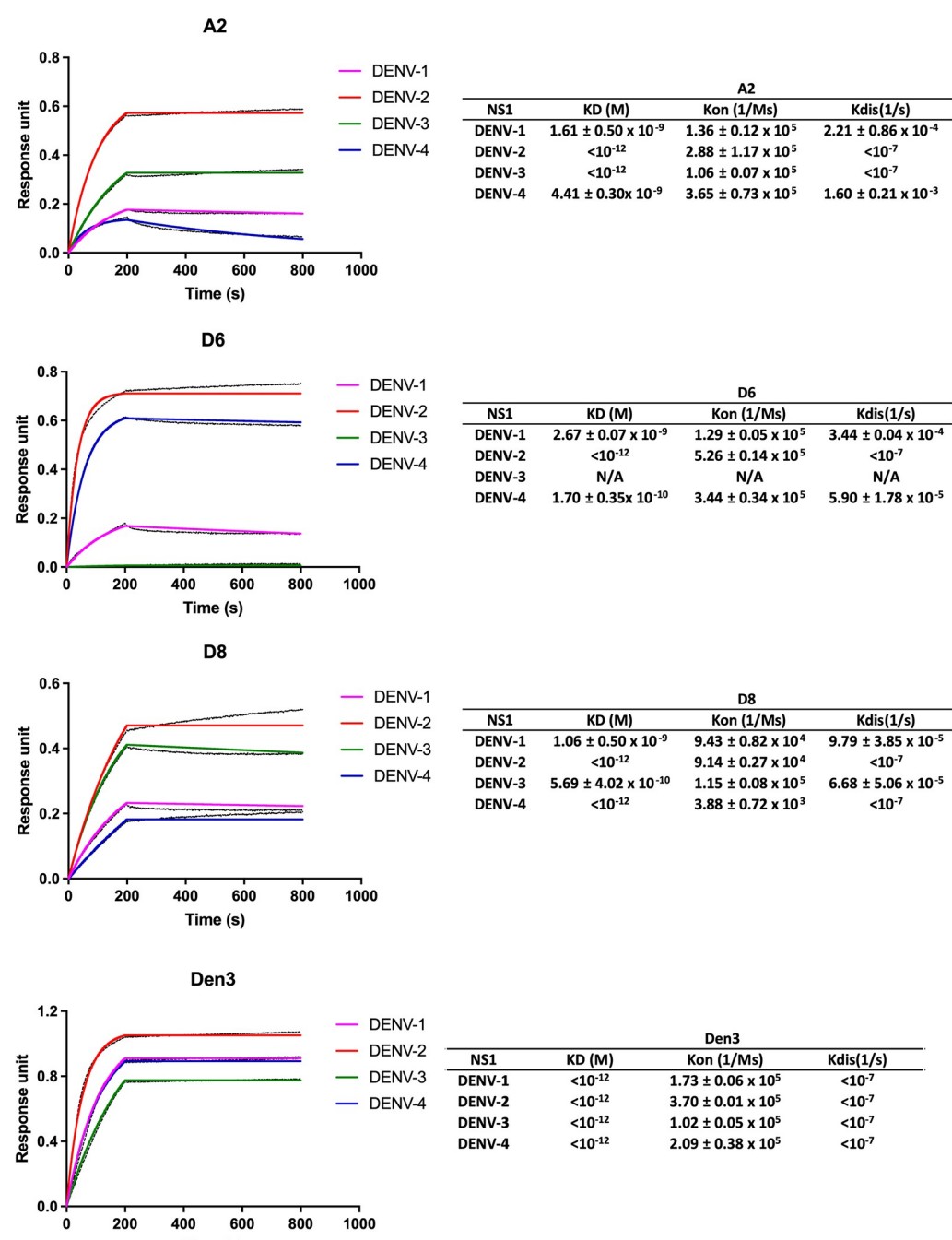

**Fig 2. Analysis of association and dissociation of NS1 from various serotypes of DENV to immobilized purified IgG by BLI.** Purified antibodies were immobilized on the biosensors, the antibodies loaded biosensors were incubated in several concentrations (0, 1.56, 3.13, 6.25, 12.5, 25, 50 and 100nM) of recombinant NS1, and then the biosensors loaded with NS1 bound to antibodies were immersed in buffer to allow for the dissociation of the NS1 from the antibodies. The graph shows raw (black line) and fitted (color lines) binding of 50 nM of NS1 from the different serotypes to immobilize purified IgG. The $K_D$, $K_{on}$ and $K_{dis}$ values were calculated using 1:1 model, global fitting analysis using Octet® Data Analysis software, and the mean and standard deviation of two independent experiments were shown.

**Table 1. Characteristics of the NS1 antibodies.** VJ assignments, CDR3 sequences, % identity.

| Ab | Heavy Chain | | | | | Light Chain | | | |
|---|---|---|---|---|---|---|---|---|---|
| | V-gene | J-gene | D-gene | CDR3 | % Identity | V-gene | J-gene | CDR3 | % Identity |
| A2 | IGHV3-30*04 or IGHV3-30-3*03 | IGHJ6*02 | IGHD5-12*01 | ARDRSDSGYDSYYYYYGMDV | 92.5 | IGLV1-51*01 | IGLJ1*01 | GSWDNSLSAYV | 94.9 |
| D6 | IGHV3-9*01 | IGHJ4*02 | IGHD3-22*01 | AKISNGYRPDN | 96.6 | IGLV3-27*01 | IGLJ2*01, or IGLJ3*01 | YCAADNNVV | 96.8 |
| D8 | IGHV4-39*01 | IGHJ4*02 | IGHD5-18*01, IGHD5-5*01 | ARLGIQLWFAIDY | 96.0 | IGLV3-19*01 | IGLJ3*02 | YSRDSSGNLWV | 95.4 |

The germline analysis of A2, D6 and D8 were shown in Table 1 and the amino acid and nucleotide sequences of the light chain and heavy chain variable domains of the antibodies are shown in S1 Table. All three antibodies were isolated from the same patient and were derived from different VH and VL genes: IGHV3-30 and IGLV1-51 for A2, IGHV3-9 and IGLV3-27 for D6, and IGHV4-39 and IGLV3-19 for D8 (Table 1). In conclusion, we have identified three antibodies that bind to multimeric forms of NS1 from multiple DENV serotypes.

## Development of a dengue-specific NS1-capture ELISA

To develop an ELISA that captures NS1 from all dengue serotypes, we tested different combinations of antibodies as capturing and detecting antibody. We combined A2 and D6 as coating antibodies or detecting antibodies in our assays to ensure that we could detect NS1 from all four DENV serotypes (Fig 3). As shown in Fig 3A and 3B, only recombinant DENV-2 NS1 was detected when the following combination of antibodies were used: (1) A2 and D6 as capture antibodies and D8 or Den3 as detecting antibodies (Fig 3A), and (2) D8 as capture antibodies and A2 and D6 or Den3 as detecting antibodies (Fig 3B). To test an alternative approach allowing detection of all four DENV serotypes, we coated the wells with Den3 and used A2/D6, D8, or a polyclonal antibodies against ZIKV NS1 as detecting antibodies. Both A2/D6 and D8 detected recombinant NS1 from all four serotypes but not NS1 from ZIKV. D8 was more sensitive in detecting NS1 from all four DENV serotypes compared to A2/D6 (Fig 3C). To confirm the presence of ZIKV NS1 that was used in the capture ELISAs, we directly coated the NS1 from all the viruses and used polyclonal antibodies against ZIKV NS1 for detection (Fig 3D). In conclusion, we established a dengue-specific NS1 capture ELISA consisting of Den3 as capture antibody and D8 as detecting antibody that detects recombinant NS1 for all dengue serotypes, but not ZIKV NS1.

## Competition ELISA suggests an overlapping epitope for A2 and D6

Based on results from the NS1 capture ELISA (Fig 3), we hypothesized that A2 and D6 bind to overlapping epitopes on dengue NS1. To test this hypothesis and to further characterize the newly identified antibodies, we performed a competition ELISA using unlabeled and HRP-labeled antibodies. Wells were coated with DENV-2-NS1, unlabeled antibodies were added to allow for binding, washed, and then HRP-labeled antibodies were added. In the presence of unlabeled A2, only 12% of A2-HRP was detected to bind to the DENV-2-NS1 and unlabeled D6 and D8 had minimal effect on binding of A2-HRP. Unlabeled A2 and D6 inhibited HRP-D6 from binding to NS1, suggesting that A2 and D6 bound to overlapping on DENV-2-NS1, although competition was only observed in one direction. Finally, only unlabeled D8 prevented binding of HRP-D8, suggesting that D8 bound to an NS1 epitope that is distinct from the A2 and D6 epitope (Fig 4).

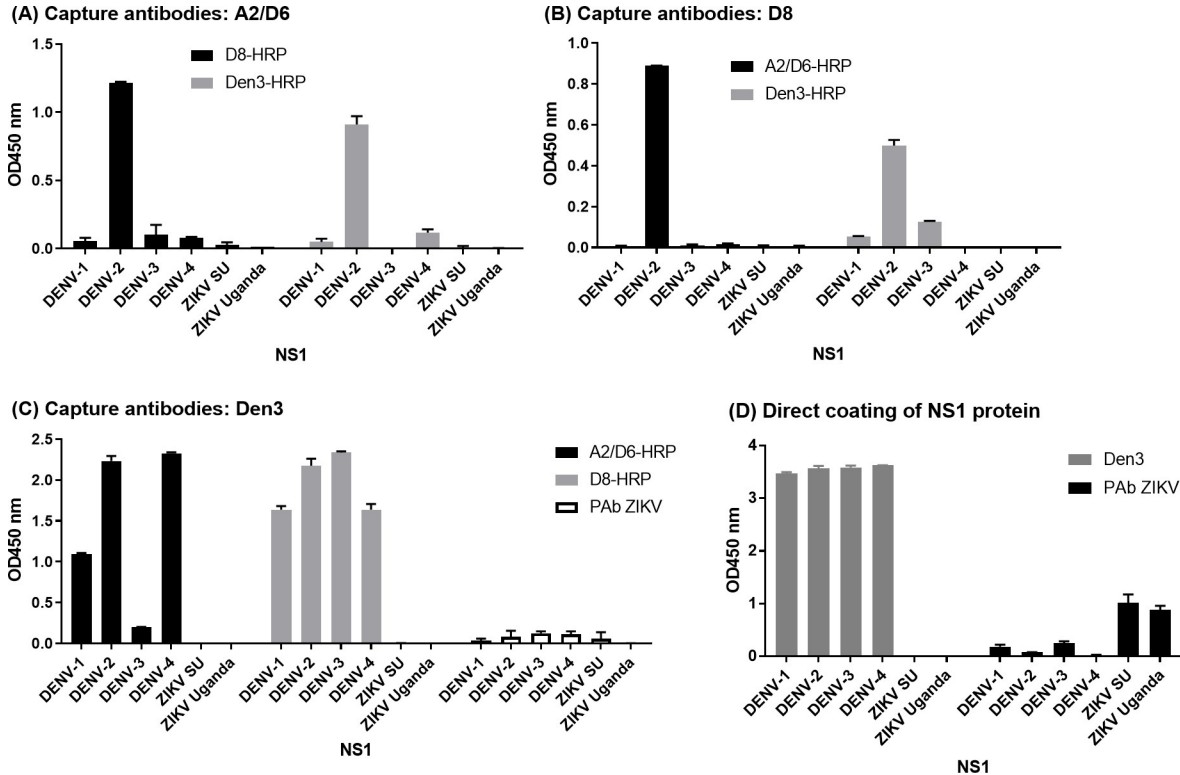

**Fig 3. NS1 capture ELISAs for the detection of NS1 from all dengue serotypes using different combinations of coating and detecting antibodies.** Wells were coated with (A) A2 and D6; (B) D8; or (C) Den3, and then purified recombinant NS1 from various viruses were added. The bound NS1 were detected by the addition of various HRP-conjugated antibodies. (D) Purified NS1 used for the sandwich ELISAs were coated directly on the wells, Den3 or PAb against ZIKV were added, and the bound antibodies were detected after the addition of HRP-conjugated anti-human or HRP-conjugated anti-rabbit. Representative results are shown and the values shown are OD± standard deviation from duplicates.

## Evaluation of the performance of the NS1 capture ELISA

To ensure that our NS1-capture ELISA that consisted of Den3 as capture antibodies and D8-HRP as detecting antibody could detect NS1 from multiple dengue serotypes and strains, we prepared NS1 by infecting Vero cells with two strains of dengue viruses from each serotype. As shown in Fig 5A, we detected NS1 from all DENV strains tested, but not ZIKV. We then determined the limit of detection of the assay using recombinant NS1 prepared in human plasma. The detection limit of the assay was defined as the concentration of NS1 with an OD value that was equal to 2-fold the mean OD value of the negative control wells. As shown in Fig 5B, the detection limits were 60 ng/mL for DENV-1, 5 ng/mL for DENV-2, 17 ng/mL for DENV-3 NS1, and 58 ng/mL for DENV-4 NS1. These results suggest that the assay was most sensitive in detecting NS1 from DENV-2, followed by NS1 from DENV-3 and DENV-4. The assay was least sensitive in detecting NS1 from DENV-1.

Finally, we examined if the ELISA could detect NS1 from dengue-infected patients during the acute phase of infection, ie. Less than or equal to 7 days after fever onset. These patients were confirmed NS1 positive using the SD. BIOLINE Dengue Duo kit that could detect NS1 antigen and IgG/IgM against NS1. Of the 60 samples tested, we detected NS1 in 9.5% (2/21) of the DENV-1 samples, 68.8% (22/32) of DENV-2 samples, 100% (6/6) of the DENV-3 samples, and 100% (2/2) of the DENV-4 samples (2/2) using the developed NS-1 capture ELISA. NS1 was not detected in samples collected from the same patient cohort at greater than 7 days after

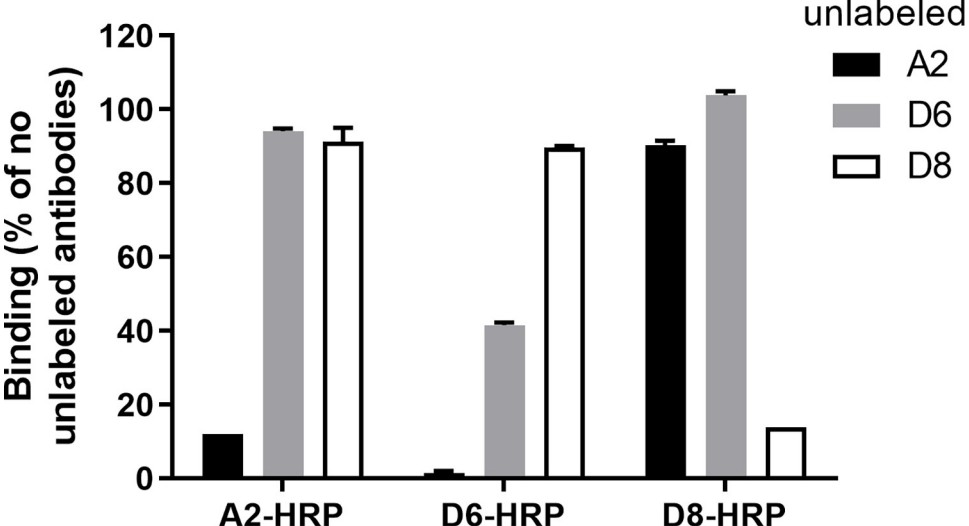

**Fig 4. Competition indirect ELISA.** Wells were coated with DENV-2 NS1, blocked, and incubated with unlabeled A2, D6, or D8. The unlabeled antibodies were washed and then A2-HRP, D6-HRP or D8-HRP were added. The bound HRP-conjugated antibodies were detected after the addition of TMB substrate. Representative results from 2 experiments are shown and the values shown are OD± standard deviation from duplicates.

fever onset demonstrating that the assay did not lead to false positive results (Fig 5C). In conclusion, the ELISA detected NS1 from dengue-infected patients.

## Discussion

In this study, we characterized three newly isolated (A2, D6, and D8) and one previously published (Den3) human antibodies against NS1 and developed a dengue NS1 capture ELISA using these antibodies. To develop a DENV-specific NS1 capture ELISA, it is important that all antibodies fulfilled the following criteria: (1) recognize NS1 from multiple dengue serotypes; (2) bind to multimeric forms of NS1; and (3) have high affinity towards NS1. We cloned antibody genes from B cells that bound to NS1 from DENV-2 and/or DENV-4 to ensure that the selected antibodies recognized NS1 from more than one dengue serotypes. Of the three isolated antibodies, both A2 and D6 recognized three out of the four serotypes, whereas D8 recognized NS1 from all four dengue serotypes, albeit at different affinity (Figs 1 and 2). Glycosylated NS1 is secreted by DENV-infected mammalian cells as hexamers [3] and can be detected in the blood of DENV-infected patients [26]. We showed by immunoblot and BLI that all our antibodies recognized multimeric forms of NS1. Finally, antibodies with high affinity towards NS1 allow better and tighter binding of the antigen to the antibodies. Of all the antibodies used in this study, Den3 has the highest affinity to NS1 from all four serotypes.

We predicted that both A2 and D6 bind to the wing and D8 binds to the β-ladder of the NS1 dimer based on serotype sequence comparison and our results (ELISA, Western blot, and BLI analysis). Based on our results, D6 interacted well to NS1 from DENV-1, -2 and -4 but not DENV-3, suggesting that D6 binds to epitopes that are conserved among DENV-1, -2, and -4. We therefore narrowed down potential binding sites for D6 to amino acids 61–71, 132–138, and 183–190, and all these amino acids are located at the wing domain of NS1 [27]. Using similar rationale, we speculated that the potential binding sites for A2 included amino acids 154–171, 252–263, and 310–323, and the potential binding sites for D8 are located at amino acids 110–120 and 154–171 within the wing domain and amino acids 252–263 and 310–323 of the

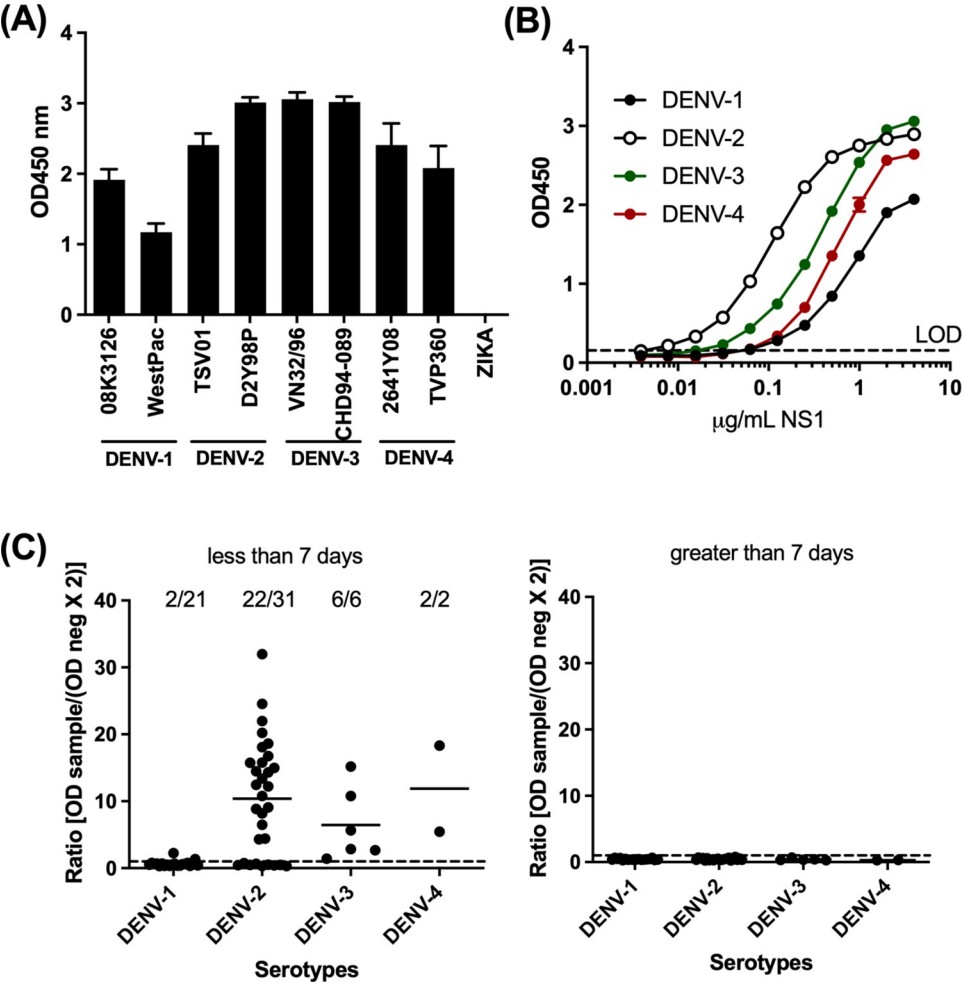

**Fig 5. Detection of NS1 by NS1-capture ELISA that consisted of Den3 as capture antibody and D8-HRP as detecting antibody.** Assay was performed using (A) supernatant from virus infected cells, (B) human plasma spiked with purified NS1 from different dengue serotypes, and (C) human plasma from patients infected with dengue viruses from ≤7 days from fever onset or >7 days from fever onset. (A) The mean OD values ± standard deviation from duplicates are shown. (B) The dotted line represents the limit of detection (LOD) of the assay that was derived from 2-fold the mean OD values of the wells containing no NS1. (C) Individual dots represent samples from different patients and the dots on and above the dotted lines are considered positive for NS1.

β-ladder domain (Fig 6). We showed by competition ELISA that A2 and D6 recognized overlapping epitopes or epitopes that are close in proximity. We, therefore, propose that A2 binds to amino acids 154–171 (wing domain) whereas D8 binds to amino acid 252–263 or 310–323 (β-ladder domain). Future studies including mutagenesis and solving the structure of the antibody-NS1 complex are needed to test our hypothesis.

The NS1-capture ELISA developed in this study specifically detected NS1 from DENV, and not ZIKV. This assay will need to be tested for the possibility to recognize NS1 from other closely related flaviviruses, such as JEV and WNV, to confirm that the assay is DENV-specific. Several commercial tests to detect for DENV NS1 in the serum are available. Lima and coworkers showed that the Platelia™ Dengue NS1 Ag ELISA did not cross-react with ZIKV NS1 using clinical samples from arbovirus suspected cases [29]. In a different study using supernatant from virus-infected cell culture, Tan and coworkers found that SD BIOLINE Dengue NS1 Ag rapid test and Panbio Dengue Early Rapid test could only detect ZIKV NS1 when

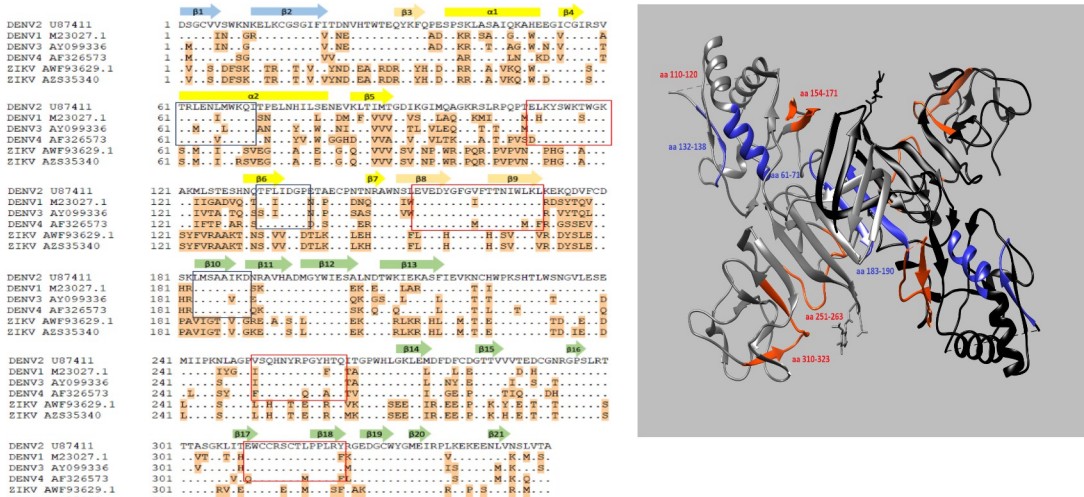

**Fig 6.** (A) Alignment of NS1 amino acid sequences from DENV and ZIKV viruses. Blue boxes are potential epitopes recognized by D6 and red boxes are potential epitopes recognized by A2 and D8. (B) DENV-2 NS1 dimer structure (PDB: 4O6B) prepared using UCSF Chimera [28]. NS1 dimer with one subunit in grey and another subunit in black. Potential epitopes recognized by D6 are colored in blue and the potential epitopes recognized by A2 and D8 are colored in red.

the virus titers were above $10^6$ PFU/mL [30]. It is important to note that the amount of ZIKV NS1 in the virus supernatant tested in their experiments was not known. We therefore do not know the sensitivity of these assays towards ZIKV NS1. One of the limitations of our study is that we did not have clinical samples from ZIKV infected patients to test the cross-reactivity of our assay to ZIKV NS1. To overcome this limitation, we have shown that our capture ELISA did not detect up to 2 μg/mL of purified recombinant ZIKV NS1 (Fig 3) and concentrated supernatant from ZIKV infected cell culture (Fig 5A).

Although our NS1 capture ELISA detected NS1 from all the dengue serotypes and strains tested in this study, we were only detecting NS1 in 9.5% of the DENV-1 patient serum and 68.8% of the DENV-2 patient serum. One explanation could be that the patient serum contained NS1 below the limit of detection of our assay at the time of sample collection. It is important to note that these patients were previously confirmed NS1 positive by the detection of NS1 antigen and/or IgG/IgM against NS1 using SD BIOLINE Dengue Duo kit [23]. The presence of antibodies against NS1 in the serum from previous infection could interfere with our assay. Alternatively, it is possible that our assay was not sensitive enough and further optimization of the assay is required to increase the limit of detection of the assay. The level of NS1 in the plasma of patients ranges from several ng/mL to μg/mL [26]. The limit of detection of our NS1 capture assay was 5 ng/mL for DENV-2 NS1, suggesting that our assay is sensitive enough to detect NS1 from DENV-2 infected patients. The limit of detection of the commercially available tests is unknown. However, similar to the commercially available tests [9–12], our assay is less sensitive in detecting NS1 from other dengue serotypes, specifically DENV-1. This result is consistent with the lower $K_D$ value of D8 towards DENV-1 NS1 (Fig 2). It is therefore important that further optimization is performed to increase the limit of detection of the assay towards other dengue serotypes. Several approaches could be used to improve the sensitivity of the assay towards other dengue serotypes. For example, a biotinylated D8 antibody could be used as a detection antibody, followed by horse-radish peroxidase conjugated streptavidin. This method has been shown to significantly increase the sensitivity of an ELISA. Other parameters that we could evaluate include different blocking buffers, the concentrations

of the coating antibodies and detecting antibodies, incubation times, incubation temperature, and different substrates.

In conclusion, we characterized three newly isolated antibodies against NS1 (A2, D6, and D8) and a previously published antibody (Den3). Using these antibodies, we have established a dengue-specific capture ELISA.

## Supporting information

**S1 Fig. B cell sorting strategy for patient samples.** Peripheral blood mononuclear cells were incubated with a cocktail of antibodies, and fluorescently labelled recombinant DENV2- and DENV4-NS1. Memory B cells that were positive for DENV2-NS1 or DENV2-NS1 and DENV4-NS1 were sorted into 96-well PCR plates.
(TIF)

**S1 Table. Nucleotide and amino acid sequences of the CDR of heavy and light chains of A2, D6, and D8.**
(DOCX)

**S1 Raw images. Original western blots and SDS-PAGE images.** These blots and gel images were cropped, compiled, and labeled to generate Fig 1B.
(PDF)

## Acknowledgments

We thank Ying-Xiu Toh and Sumathy Velumani for technical support and SIgN flow cytometry for sorting the B cells. Molecular graphics was performed with UCSF Chimera developed by the Resource for Biocomputing, Visualization, and Informatics at the University of California, San Francisco, with support from NIH P41-GM103311.

## Author Contributions

**Conceptualization:** Pei-Yin Lim, Appanna Ramapraba, Katja Fink.

**Data curation:** Pei-Yin Lim, Appanna Ramapraba, Thomas Loy, Angeline Rouers, Tun-Linn Thein, Yee-Sin Leo.

**Formal analysis:** Pei-Yin Lim, Appanna Ramapraba, Thomas Loy, Angeline Rouers, Tun-Linn Thein.

**Funding acquisition:** Katja Fink.

**Investigation:** Pei-Yin Lim.

**Methodology:** Pei-Yin Lim, Appanna Ramapraba, Thomas Loy, Angeline Rouers, Tun-Linn Thein, Yee-Sin Leo.

**Resources:** Dennis R. Burton.

**Supervision:** Pei-Yin Lim, Katja Fink, Cheng-I Wang.

**Validation:** Pei-Yin Lim.

**Writing – original draft:** Pei-Yin Lim.

**Writing – review & editing:** Pei-Yin Lim, Appanna Ramapraba, Thomas Loy, Angeline Rouers, Tun-Linn Thein, Yee-Sin Leo, Dennis R. Burton, Katja Fink, Cheng-I Wang.

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
