## [Decision Letter · Decision Letter 0]

5 Oct 2022

PONE-D-22-24631A nonstructural protein 1 capture enzyme-linked immunosorbent assay specific for dengue virusesPLOS ONE

Dear Dr. Lim,

Thank you for submitting your manuscript to PLOS ONE. After careful consideration, we feel that it has merit but does not fully meet PLOS ONE’s publication criteria as it currently stands. Therefore, we invite you to submit a revised version of the manuscript that addresses the points raised during the review process.

Both reviewers identified several areas for improvement including further explanation of key experiments and missing methodological information. Please carefully address their comments, especially where they have asked for explanations of potentially discrepant results.

We look forward to receiving your revised manuscript.

Kind regards,

Kevin A. Henry

Academic Editor

PLOS ONE

Journal Requirements:

"This work was supported by SIgN Core grant"

"The authors receive no specific funding for this work."

Reviewers' comments:

Reviewer's Responses to Questions

**Comments to the Author**

1. Is the manuscript technically sound, and do the data support the conclusions?

Reviewer #1: Partly

Reviewer #2: Yes

2. Has the statistical analysis been performed appropriately and rigorously? 

Reviewer #1: I Don't Know

Reviewer #2: N/A

3. Have the authors made all data underlying the findings in their manuscript fully available?

Reviewer #1: Yes

Reviewer #2: Yes

4. Is the manuscript presented in an intelligible fashion and written in standard English?

Reviewer #1: Yes

Reviewer #2: Yes

5. Review Comments to the Author

Reviewer #1: The authors developed and characterized three new human monoclonal antibodies that bind to DENV NS1 but not to ZIKV NS1. Using the three new antibodies and a previously described antibody that binds to NS1 from each of the four DENV serotypes the authors devised ELISAs and found a combination of antibodies that were able to detect recombinant NS1 from all four serotypes as well as NS1 in viral supernatants from the four serotypes but did not show signal for ZIKV. Finally, the authors tested the ability of their ELISA using patient samples.

1. In the introduction the authors state that Yellow fever virus, West Nile virus, and Japanese encephalitis virus are also closely related to DENV. Is it important to ensure no cross reactivity with these viruses?

2. Why does the signal continue to go higher during the dissociation phage of the BLI data? Does this make sense? Please offer some explanation about the BLI curves

3. Please explain why A2 has KD of 4.4 nM to DENV-4 NS1 but does not bind to DENV-4 in the indirect ELISA

4. In Figure 5B the developed ELISA shows the best detection of the recombinant DENV 2 NS1. In addition, the D8 antibody was isolated from a convalescent DENV-2 patient. However, 5C shows that about 1/3 of the DENV-2 patient samples are not detected by the author’s ELISA. How could the ELISA be improved to better detect DENV-2 in patient samples as well as to detect DENV-1 in patient samples.

5. It is very unclear how the authors narrowed down the areas of the potential epitopes. Please explain in more detail.

Reviewer #2: This paper describes the isolation of anti-Dengue NS1 antibodies from a Dengue patient, using B-cell sorting. The purpose was to find new reagents to use in Dengue diagnostic ELISA kits which are specific for Dengue NS1 with no cross-reactivity to other flaviviruses such as ZIKA. The authors screened for antibodies which would bind to all four Dengue serotypes, and isolated one pan-specific antibody (D8). When paired with a published pan-specific antibody (Den3) a sandwich ELISA to detect Dengue NS1 was developed. The authors also speculated on the epitope of the D8 and Den3 antibodies, as well as other isolated antibodies which bound one or more Dengue serotypes, by performed competition ELISAs.

General comments:

The manuscript is well written and organised.

It would be good to include more detail on the isolation of the antibodies, and include some results (eg show the flow cytometry gating strategy, and a typical flow cytometry result of a positive hit). How many B-cells were screened, and what was the positive hit rate?

Since the purpose of the study was to find alternative reagents for diagnostic NS1 detection ELISA, it would be good to see a side-by-side comparison of the developed ELISA with an existing commercial kit, especially for limit of detection with recombinant Dengue and ZIKA NS1.

Specific comments:

Line 68: How many patients?

Line 81: Describe sorting/gating conditions

Line 86: Reference 14 is cited for the detailed method for cloning and expressing the antibodies. Please check that the correct reference has been cited, as Ref14 does not contain this methodology

Line 87: Primer sequences? Or are they described in the correct reference 14?

Line 129: This is the first time Den3 antibody has been mentioned (other than the abstract). In the abstract it is referred to as previously published. Please cite the publication here – it is later cited in the results. Also how did you obtain this antibody – from the authors of Ref21? Or expressed from a published sequence – if so, where did the sequence come from? Ref 21 uses the Den3 antibody, but doesn’t give any information about its source – is there a better reference?

Line 159: The method suggests different concentrations of NS1 were included in the analysis, but the results only show one concentration (50nM). Please clarify if multiple concentrations were used, and if so, were these included in the analysis to give more accurate results?

Line 160: Describe evaluation for BLI analysis (fit to a 1:1 binding model?)

Line 288: The word ‘and’ should be ‘an’

Lines 304-311: The samples were confirmed positive using the Bioline Dengue kit. For the samples that were not detected by the newly developed assay, do you think these are false negatives in the new assay or false positives in the Bioline assay? As the new assay was the least sensitive for DENV1, this could explain the low detection of DENV1 samples; however the detection of DENV2 was also fairly low when the assay has the highest sensitivity for DENV2

6. PLOS authors have the option to publish the peer review history of their article (what does this mean?). If published, this will include your full peer review and any attached files.

Reviewer #1: No

Reviewer #2: No

---

## [Author Response · Author response to Decision Letter 0]

17 Nov 2022

The response to reviewers and editor comments is included in the rebuttal letter. Please refer to the rebuttal letter as it contains graphs to support the explanation. 

Reviewer #1: The authors developed and characterized three new human monoclonal antibodies that bind to DENV NS1 but not to ZIKV NS1. Using the three new antibodies and a previously described antibody that binds to NS1 from each of the four DENV serotypes the authors devised ELISAs and found a combination of antibodies that were able to detect recombinant NS1 from all four serotypes as well as NS1 in viral supernatants from the four serotypes but did not show signal for ZIKV. Finally, the authors tested the ability of their ELISA using patient samples.

1. In the introduction the authors state that Yellow fever virus, West Nile virus, and Japanese encephalitis virus are also closely related to DENV. Is it important to ensure no cross reactivity with these viruses?

Response: Yellow fever virus, West Nile virus, Japanese encephalitis, Dengue viruses, and Zika viruses are considered closely related because they are all members of the Flaviviridae family. These viruses are classified based on their biological properties such as virus morphology, genome structures or antigenic properties. There is no concern about cross reactivity of DENV NS1 with Yellow fever virus, West Nile virus, and Japanese encephalitis virus in the field because the sequence homology of DENV NS1 with these viruses are less than 50% (Xu et. al., 2016).

2. Why does the signal continue to go higher during the dissociation phage of the BLI data? Does this make sense? Please offer some explanation about the BLI curves

Response: In Biolayer interferometry (BLI), the mAbs were immobilized onto a biosensor tip, binding of the antigen (NS1) to the mAb led to an increase of optical thickness, resulting in a wavelength shift. This technique is sensitive and may have a slight deviation due to background, explaining why a slight increase can be observed during the reading for the dissociation step. Fig 2 (in our initially submitted manuscript) represents raw data (not fitted), while the analysis to determine the KD are based on fitted data. For clarity, we have updated Fig 2 in the revised manuscript to show the fitted data, instead of the raw data.

3. Please explain why A2 has KD of 4.4 nM to DENV-4 NS1 but does not bind to DENV-4 in the indirect ELISA

Response: A2 has a KD of 4.4nM to DENV-4 NS1 as measured by BLI but no detectable A2 binding to DENV-4 NS1 was observed by indirect ELISA. This is because indirect ELISA is in general a less sensitive method as compared to BLI due to the differences in technology and means of measuring interactions between antibody and antigen.

BLI is a very sensitive technique with a system limit of detection for KD measurements between 1mM-1pM and the sensitivity of the indirect ELISA depends upon the ELISA. In addition, BLI involves immobilizing the antibodies through its Fc domain on the biosensor tip, and then allowing soluble NS1 to bind in solution. Binding of antigen to the immobilized antibodies causes an increase in optical thickness at the biosensor tip, resulting in a wavelength shift proportional to the extent of binding.

Indirect ELISA, however, involves immobilizing soluble NS1 onto the surface, allowing antibodies to bind to the immobilized NS1, and then the bound antibodies were detected using an HRP-conjugated secondary antibodies and a colorimetric substrate. As the NS1 could be immobilized onto the surface via different orientations, some of these epitopes may not be accessible to the antibodies. This could also contribute to one of the reasons why ELISA may be less sensitive as compared to BLI.

4. In Figure 5B the developed ELISA shows the best detection of the recombinant DENV 2 NS1. In addition, the D8 antibody was isolated from a convalescent DENV-2 patient. However, 5C shows that about 1/3 of the DENV-2 patient samples are not detected by the author’s ELISA. How could the ELISA be improved to better detect DENV-2 in patient samples as well as to detect DENV-1 in patient samples.

Response: There are several ways to improve the sensitivity of the assay. One way is to use a biotinylated-D8 antibody as a detection antibody, followed by horse-radish peroxidase conjugated streptavidin. This method has been shown to significantly increase the sensitivity of an assay. In addition, we could optimize the concentrations of the coating antibodies, the biotinylated-D8 antibody and horse-radish peroxidase conjugated streptavidin. Other parameters that we could examine includes incubation times, incubation temperature, and different TMB substrates.

5. It is very unclear how the authors narrowed down the areas of the potential epitopes. Please explain in more detail.

Response: We have revised the discussion section to further describe how we have speculated the potential epitopes recognized by the antibodies.

Reviewer #2: This paper describes the isolation of anti-Dengue NS1 antibodies from a Dengue patient, using B-cell sorting. The purpose was to find new reagents to use in Dengue diagnostic ELISA kits which are specific for Dengue NS1 with no cross-reactivity to other flaviviruses such as ZIKA. The authors screened for antibodies which would bind to all four Dengue serotypes, and isolated one pan-specific antibody (D8). When paired with a published pan-specific antibody (Den3) a sandwich ELISA to detect Dengue NS1 was developed. The authors also speculated on the epitope of the D8 and Den3 antibodies, as well as other isolated antibodies which bound one or more Dengue serotypes, by performed competition ELISAs.

General comments:

The manuscript is well written and organised.

It would be good to include more detail on the isolation of the antibodies, and include some results (eg show the flow cytometry gating strategy, and a typical flow cytometry result of a positive hit). How many B-cells were screened, and what was the positive hit rate?

Since the purpose of the study was to find alternative reagents for diagnostic NS1 detection ELISA, it would be good to see a side-by-side comparison of the developed ELISA with an existing commercial kit, especially for limit of detection with recombinant Dengue and ZIKA NS1.

Specific comments:

Line 68: How many patients?

Response: We have updated the method and material section to indicate that the antibodies were isolated from one patient.

Line 81: Describe sorting/gating conditions

Response: We have added the sorting/gating strategy as S1 Fig and added the description of the strategy in the results section.

Line 86: Reference 14 is cited for the detailed method for cloning and expressing the antibodies. Please check that the correct reference has been cited, as Ref14 does not contain this methodology

Response: We have checked and confirmed that the correct reference (Appanna et. al., 2016; originally Ref14) section 2.3 described the detailed method for cloning and expressing the antibodies.

Line 87: Primer sequences? Or are they described in the correct reference 14?

Response: We have confirmed that the primer sequences are also listed in table S1 in the publication Appanna et. al., 2016 (originally Reference 14).

Line 129: This is the first time Den3 antibody has been mentioned (other than the abstract). In the abstract it is referred to as previously published. Please cite the publication here – it is later cited in the results. Also how did you obtain this antibody – from the authors of Ref21? Or expressed from a published sequence – if so, where did the sequence come from? Ref 21 uses the Den3 antibody, but doesn’t give any information about its source – is there a better reference?

Response: We have added the reference as per reviewer’s suggestions (currently Line 147). Cells expressing Den3 was obtained from one of the coauthors, Dennis Burton and this antibody was first published in Ref21. We have added a section in the Methods and Materials to describe the production and purification of Den3.

Line 159: The method suggests different concentrations of NS1 were included in the analysis, but the results only show one concentration (50nM). Please clarify if multiple concentrations were used, and if so, were these included in the analysis to give more accurate results?

Response: Several concentrations of NS1 were used in the assay: 1.56, 3.13, 6.25, 12.5, 25, 50 and 100 nM. In general, graph from one concentration is demonstrated in most publications. We have chosen to present only the graph for 50nM as a representation but the KD values shown in the table were calculated by taking all the concentrations into account (global fitting analysis).

Thus, the value presented in the table on the right of Fig 2 include the different concentrations for the calculation in order to deliver more accurate results. 

We have added additional information in the Method and Material, the legend for Figure 2, and the Results sections to clarify the analysis.

Line 160: Describe evaluation for BLI analysis (fit to a 1:1 binding model?)

Response: We have added the evaluation of BLI analysis in the Methods and Materials section. We used the Octet® Data Analysis software for affinity calculations as detailed in Noy-Porat et. al., 2021. In short, we used the condition without NS1 antigen (0 nM) as background reference to subtract to the data. Y axis was aligned to the baseline (between 190 and 199.8 sec which was consider the most stable). The association and dissociation steps were analysed using 1:1 model with global fitting. 

Line 288: The word ‘and’ should be ‘an’

Response: We have edited the “and” to “an”.

Lines 304-311: The samples were confirmed positive using the Bioline Dengue kit. For the samples that were not detected by the newly developed assay, do you think these are false negatives in the new assay or false positives in the Bioline assay? As the new assay was the least sensitive for DENV1, this could explain the low detection of DENV1 samples; however the detection of DENV2 was also fairly low when the assay has the highest sensitivity for DENV2

Response: We are unable to conclude whether or not the samples that were positive by Bioline Dengue kit but not the newly developed assay were false negatives in the assay or false positive in the Bioline assay for the following reasons: (1) the newly developed assay could be optimized further to increase sensitivity; and (2) SD Bioline Dengue kit detects both NS1 antigen and IgG/IgM against NS1, whereas the newly developed assay only measures NS1 antigen; therefore, SD Bioling Dengue kit could detect more positive samples. We have revised the results section to clarify that Bioline Dengue kit could detect both NS1 antigen and antibodies against NS1.

---

## [Decision Letter · Decision Letter 1]

4 Dec 2022

PONE-D-22-24631R1A nonstructural protein 1 capture enzyme-linked immunosorbent assay specific for dengue virusesPLOS ONE

Dear Dr. Lim,

Thank you for submitting your manuscript to PLOS ONE. After careful consideration, we feel that it has merit but does not fully meet PLOS ONE’s publication criteria as it currently stands. Therefore, we invite you to submit a revised version of the manuscript that addresses the points raised during the review process. Both reviewers appreciated the changes and revisions made in light of their initial comments, but felt that a small number of issues still remained to be addressed prior to publication. See their detailed comments.

We look forward to receiving your revised manuscript.

Kind regards,

Kevin A. Henry

Academic Editor

PLOS ONE

Journal Requirements:

Reviewers' comments:

Reviewer's Responses to Questions

**Comments to the Author**

1. If the authors have adequately addressed your comments raised in a previous round of review and you feel that this manuscript is now acceptable for publication, you may indicate that here to bypass the “Comments to the Author” section, enter your conflict of interest statement in the “Confidential to Editor” section, and submit your "Accept" recommendation.

Reviewer #1: (No Response)

Reviewer #2: (No Response)

2. Is the manuscript technically sound, and do the data support the conclusions?

Reviewer #1: Partly

Reviewer #2: Yes

3. Has the statistical analysis been performed appropriately and rigorously? 

Reviewer #1: I Don't Know

Reviewer #2: N/A

4. Have the authors made all data underlying the findings in their manuscript fully available?

Reviewer #1: Yes

Reviewer #2: Yes

5. Is the manuscript presented in an intelligible fashion and written in standard English?

Reviewer #1: Yes

Reviewer #2: Yes

6. Review Comments to the Author

Reviewer #1: 1. Previously I had made the comment: “In the introduction the authors state that Yellow fever virus, West Nile virus, and Japanese encephalitis virus are also closely related to DENV. Is it important to ensure no cross reactivity with these viruses?” In their response to my comment, the authors stated that “There is no concern about cross reactivity of DENV NS1 with Yellow fever virus, West Nile virus, and Japanese encephalitis virus in the field because the sequence homology of DENV NS1 with these viruses are less than 50% (Xu et. al., 2016)”. However, according to table 2 of that reference, homology to JEV ranges from 51.1% to 53.4%, homology to WNV ranges from 50.3% to 55.4%, and homology to ZIKV ranges from 53.4% to 55.1%. Only YFV has less than 50% homology to the four DENV serotypes.

In order to have the conclusions of this paper be justified by the data, please include a statement indicating that further testing for binding to NS1 from the related WNV, YFV, and JEV will need to be conducted to definitively show the specificity of the developed assay.

2. In Figure 5B the developed ELISA shows the best detection of the recombinant DENV 2 NS1 with a limit of detection of 5 ng/mL. However, 5C shows that about 30% of the DENV-2 patient samples are not detected by the author’s ELISA. Was this expected? Please add a couple of sentences to the discussion section of the manuscript about these results which can include next steps to potentially improve the assay.

3. Please check the size of boxes in figure 6. There is one box that includes one of the ZIKV sequences when they probably all are meant to only include the four DENV sequences.

Reviewer #2: In general, the authors have responded to both reviewer’s comments adequately. However, the following requires further clarification or changes to the manuscript:

Responses to Reviewer 1’s comments:

2. In response to Reviewer 1’s comment about the BLI results, the authors have changed the figure to show the fitted data rather than the raw data. This just hides the issue that the reviewer was concerned about (ie the upward drifting dissociation phase). The graphs should show both the raw data and the fitted curves (eg in solid and dotted lines for each concentration). The upward drift is likely caused by non-specific binding of the analyte to the reference sensor. Although the data has subtracted the 0nM data during the analysis, was there also a reference sensor subtraction for each concentration of analysis (ie binding of the NS1 to a sensor without antibody)?

4. The authors have responded to Reviewer 1’s comments with suggestions on how to optimise the ELISA to improve the sensitivity. Have the authors tried these relatively simple measures and can they be incorporated into the manuscript? If this is not possible, then some comment in the discussion should be included to state that improved sensitivity is required since the ELISA did not detect some of the positive patient samples, and discuss these methods.

Responses to Reviewer 2:

The authors have not commented on Reviewer 2’s suggestion to include a side-by-side comparison with an existing commercial kit. If the purpose is to find reagents to develop a superior assay then this comparison should be included. A comparison of the limit of detection would be ideal.

Line 122: Since Dr Dennis Burton is a co-author, you do not need to acknowledge that you got the cells from Dr Burton. Instead, change the wording to: ‘Chinese hamster ovary cells expressing Den3 antibodies (ref 16) were expressed in glutamine-free…………’

7. PLOS authors have the option to publish the peer review history of their article (what does this mean?). If published, this will include your full peer review and any attached files.

Reviewer #1: No

Reviewer #2: No

---

## [Author Response · Author response to Decision Letter 1]

21 Apr 2023

Reviewer #1: 1. Previously I had made the comment: “In the introduction the authors state that Yellow fever virus, West Nile virus, and Japanese encephalitis virus are also closely related to DENV. Is it important to ensure no cross reactivity with these viruses?” In their response to my comment, the authors stated that “There is no concern about cross reactivity of DENV NS1 with Yellow fever virus, West Nile virus, and Japanese encephalitis virus in the field because the sequence homology of DENV NS1 with these viruses are less than 50% (Xu et. al., 2016)”. However, according to table 2 of that reference, homology to JEV ranges from 51.1% to 53.4%, homology to WNV ranges from 50.3% to 55.4%, and homology to ZIKV ranges from 53.4% to 55.1%. Only YFV has less than 50% homology to the four DENV serotypes.

In order to have the conclusions of this paper be justified by the data, please include a statement indicating that further testing for binding to NS1 from the related WNV, YFV, and JEV will need to be conducted to definitively show the specificity of the developed assay.

Response: We have added the statement as suggested by the reviewer (p17).

2. In Figure 5B the developed ELISA shows the best detection of the recombinant DENV 2 NS1 with a limit of detection of 5 ng/mL. However, 5C shows that about 30% of the DENV-2 patient samples are not detected by the author’s ELISA. Was this expected? Please add a couple of sentences to the discussion section of the manuscript about these results which can include next steps to potentially improve the assay.

Response: We have included discussion of these results in the discussion section (p18). As described in the discussion, the patients were previously confirmed NS1 positive using SD BIOLINE Dengue Duo kit that detects Dengue NS1 antigen and/or antibodies in the serum. It is possible that the 30% DENV-2 patient samples contained NS1 below the limit of detection of our assay. Alternatively, these patient samples may contain antibodies against NS1 from previous infection that interfere with our assay.

3. Please check the size of boxes in figure 6. There is one box that includes one of the ZIKV sequences when they probably all are meant to only include the four DENV sequences.

Response: We have amended the Figure to include only the DENV sequences.

Reviewer #2: In general, the authors have responded to both reviewer’s comments adequately. However, the following requires further clarification or changes to the manuscript:

Responses to Reviewer 1’s comments:

2. In response to Reviewer 1’s comment about the BLI results, the authors have changed the figure to show the fitted data rather than the raw data. This just hides the issue that the reviewer was concerned about (ie the upward drifting dissociation phase). The graphs should show both the raw data and the fitted curves (eg in solid and dotted lines for each concentration). The upward drift is likely caused by non-specific binding of the analyte to the reference sensor. Although the data has subtracted the 0nM data during the analysis, was there also a reference sensor subtraction for each concentration of analysis (ie binding of the NS1 to a sensor without antibody)?

Response: We only used the 0nM condition for subtraction of background. We understand the reviewer’s concern and have added both raw data and fitted data in the graphs. 

We are unable to explain the upward drifting during the dissociation phase. However, we would like to add the following points that argue against non-specific binding of the analyte to the reference sensor for the following reasons: (1) This upward drifting was only observed in antibodies with high affinity to NS1. Specifically, DENV2 and DENV4 NS1 for A2, DENV2 NS1 for D6, and DENV2 and DENV4 for D8; (2) The experiment has been repeated twice and consistent results were observed; (3) the dissociation phase occurred in a buffer with no antigen; and (4) new biosensor tips were used for all experiments and the tips were not regenerated. We have added a few sentences to describe the observation (p11).

4. The authors have responded to Reviewer 1’s comments with suggestions on how to optimise the ELISA to improve the sensitivity. Have the authors tried these relatively simple measures and can they be incorporated into the manuscript? If this is not possible, then some comment in the discussion should be included to state that improved sensitivity is required since the ELISA did not detect some of the positive patient samples, and discuss these methods.

Response: We have added some comments in the discussion (p18).

Responses to Reviewer 2:

The authors have not commented on Reviewer 2’s suggestion to include a side-by-side comparison with an existing commercial kit. If the purpose is to find reagents to develop a superior assay then this comparison should be included. A comparison of the limit of detection would be ideal.

Response: We acknowledge the reviewer’s suggestion in performing a side-by-side comparison with an existing commercial kit. We would like to emphasize that the goal of this paper is to develop an assay that specifically detect NS1 from dengue viruses and not detect NS1 from Zika virus, and limited studies were performed to examine the specificity of the commercial kits. In addition, our assay needs further optimization before a benchmarking with commercial kits would be conducted. Finally, it is challenging to perform a side-by-side comparison for the concentration of NS1 with an existing commercial kit because commercial kits, at least those we are aware of, are not quantitative tests. The read out of a commercial kit is an OD ratio and there is no standard curve where one could interpolate the amount of NS1 from the standard curve. 

Line 122: Since Dr Dennis Burton is a co-author, you do not need to acknowledge that you got the cells from Dr Burton. Instead, change the wording to: ‘Chinese hamster ovary cells expressing Den3 antibodies (ref 16) were expressed in glutamine-free…………’

Response: We have updated the text accordingly.

---

## [Editor Report · Decision Letter 2]

4 May 2023

A nonstructural protein 1 capture enzyme-linked immunosorbent assay specific for dengue viruses

PONE-D-22-24631R2

Dear Dr. Wang,

We’re pleased to inform you that your manuscript has been judged scientifically suitable for publication and will be formally accepted for publication once it meets all outstanding technical requirements.

Kind regards,

Kevin A. Henry

Academic Editor

PLOS ONE
---

## [Editor Report · Acceptance letter]

9 May 2023

PONE-D-22-24631R2 

A nonstructural protein 1 capture enzyme-linked immunosorbent assay specific for dengue viruses 

Dear Dr. Wang:

I'm pleased to inform you that your manuscript has been deemed suitable for publication in PLOS ONE. Congratulations! Your manuscript is now with our production department. 

Kind regards, 

on behalf of

Dr. Kevin A. Henry 

Academic Editor

PLOS ONE